

# Cyber-attack method and perpetrator prediction using machine learning algorithms

Abdulkadir Bilen[1] and Ahmet Bedri Özer[2]

[1] Criminal Department, General Directorate of Security, Ankara, Turkey
[2] Department of Computer Engineering, Firat University, Elazığ, Turkey

## ABSTRACT

Cyber-attacks have become one of the biggest problems of the world. They cause serious financial damages to countries and people every day. The increase in cyber-attacks also brings along cyber-crime. The key factors in the fight against crime and criminals are identifying the perpetrators of cyber-crime and understanding the methods of attack. Detecting and avoiding cyber-attacks are difficult tasks. However, researchers have recently been solving these problems by developing security models and making predictions through artificial intelligence methods. A high number of methods of crime prediction are available in the literature. On the other hand, they suffer from a deficiency in predicting cyber-crime and cyber-attack methods. This problem can be tackled by identifying an attack and the perpetrator of such attack, using actual data. The data include the type of crime, gender of perpetrator, damage and methods of attack. The data can be acquired from the applications of the persons who were exposed to cyber-attacks to the forensic units. In this paper, we analyze cyber-crimes in two different models with machine-learning methods and predict the effect of the defined features on the detection of the cyber-attack method and the perpetrator. We used eight machine-learning methods in our approach and concluded that their accuracy ratios were close. The Support Vector Machine Linear was found out to be the most successful in the cyber-attack method, with an accuracy rate of 95.02%. In the first model, we could predict the types of attacks that the victims were likely to be exposed to with a high accuracy. The Logistic Regression was the leading method in detecting attackers with an accuracy rate of 65.42%. In the second model, we predicted whether the perpetrators could be identified by comparing their characteristics. Our results have revealed that the probability of cyber-attack decreases as the education and income level of victim increases. We believe that cyber-crime units will use the proposed model. It will also facilitate the detection of cyber-attacks and make the fight against these attacks easier and more effective.

## INTRODUCTION

Nowadays, it has become exceedingly difficult to ensure the security of our systems including both corporate and personal data. Major countries, such as the United States and

Corresponding author
Abdulkadir Bilen,
abdulkadir.bilen@egm.gov.tr

the United Kingdom, struggle with cyber-attacks and crimes by producing various security strategies (*Reid & Van Niekerk, 2014*). Countries are striving to ensure security in cyber space and adapt to this field (*Goel, 2020*). Protecting the critical infrastructures has a vital importance for countries. Chemical, financial, health and energy sectors, even nuclear power plants in some countries can be counted among these (*CISA, 2020*). Due to millions of cyber-attacks, financial losses significantly increase day by day (*Jang-Jaccard & Nepal, 2014*). In 2020, data stolen from the information system of Airbus Company were put on the dark web market. Medical data of millions of people have been stolen and even state of emergency has been declared due to attacks on some cities (*Check Point Security Report, 2020*). The most important elements ensuring cyber security are integrity, confidentiality, authentication, authorization, nonrepudiation and availability (*Bayuk et al., 2012*).

With each passing day, the work force becomes insufficient in fighting against cyber incidents and new solutions are sought. Solutions such as autonomous cyber defense systems (*Crawford, 2017*), smart cyber security assistant architecture (*Sayan, 2017*) and intrusion detection systems (*Ben-Asher & Gonzalez, 2015*) are investigated in the fight against cyber-attacks and crimes. Researchers use machine-learning methods to detect power outages due to cyber-attacks (*Wang et al., 2019*) and to prevent vulnerabilities of the Internet of things (*Zolanvari et al., 2019*). Other areas of use are to determine spam and network attacks (*Canbek, Sagiroglu & Temizel, 2018*), to detect the phishing attacks against the banking sector (*Moorthy & Pabitha, 2020*) and to reduce sexual crimes on social media (*Ngejane et al., 2018*). These methods have been implemented in fields as stock prediction (*Gurjar et al., 2018*), risk mapping by crimes (*Wheeler & Steenbeek, 2020*) and cyber profiling (*Zulfadhilah, Prayudi & Riadi, 2016*). Predicting crime trend and pattern (*Biswas & Basak, 2019*), criminal identity detection (*Bharathi, Indrani & Prabakar, 2017*) and crime prevention (*Lin, Chen & Yu, 2017*) are also areas of implementation.

There are three main objectives in our study. The first is to use actual cyber-crime data as input to predict a cyber-crime method and compare the accuracy results. The second is to measure whether cyber-crime perpetrators can be predicted based on the available data. The third objective is to understand the effect of victim profiles on cyber-attacks.

In this paper, real cyber-crime data of 5-years in Elazığ/Turkey were used. By using machine-learning methods, the method of attack was predicted and the perpetrator was detected. The detection was based on features like age, gender, income, education, marital status, occupation and the damage of the crime. By working on certain features such as age, gender, etc., we predicted the kind of attack methods and the victims of these attacks. These results will be used in cyber-crime modelling and attack detection by the police forces dealing with cyber-crimes. Major contributions of the proposed approach are:

- Providing advantages to cyber-crime department as it allows using actual data;
- Enabling prediction of attacks that victims may be exposed to;
- Allowing determination of the optimum performance by comparing machine-learning algorithms.

In "Related Works", the literature is reviewed, and the current studies and the missing parts are revealed. "Materials and Methods" explains the machine-learning methods to be used in research. "Results and Discussion" presents the predictions and accuracy rate from the dataset and a comparison with previous studies. "Conclusions and Future Work" includes results and future work.

## RELATED WORKS

The importance of the fight against such cyber-attacks, cyber-crimes and cyber security is highlighted in various studies. Cyber security is the protection of physical-digital data, networks, and technological systems from cyber-attacks, unauthorized accesses, disruptions, modifications, destructions and damages through various processes, applications and applied technologies (*Fischer, 2009*). Cyber-attacks such as distributed denial of service attacks by sending malicious packets (*Kaur Chahal, Bhandari & Behal, 2019*), phishing attacks to banking and shopping sites that deceive the user (*Sahingoz et al., 2019*) have increased significantly. In addition, attackers have been using malicious attack software (virus, worms, trojans, spyware and ransomware) that is installed into the user's computer without any consent of the user (*Biju, Gopal & Prakash, 2019*) increasingly. Again, the most common of these attacks and one of the attacks that are most difficult to be prevented is the social engineering attacks. They are based on technical skill, cunning and persuasion, made by taking advantage of the weakness of the victim. Kevin Mitnick, one of the world's famous hackers in social engineering attacks, penetrated most systems he attacked with this method (*Mitnick & Simon, 2009*). In the work by *Breda, Barbosa & Morais (2017)* this attack is mentioned as one of the biggest security vulnerabilities in the system no matter how secure a technical system is. Likewise, attacks against IoT devices, which have increased rapidly in recent years, affect the society considerably. Thus, attacks and threats to the IoT structure should be understood for security purposes (*Kagita et al., 2020*). Studies conducted to understand and combat cyber-attacks reveal the importance of crime prediction as discussed in this study.

The attacks described above are defined as prohibited criminal acts within the legal framework of many governments. The duty of fighting against crime and criminals is given to law enforcement departments. Researchers assist the institutions conducting the investigation with various analysis and prediction methods. For example, big data (*Rewari & Singh, 2017*) and machine-learning (*Lin, Chen & Yu, 2017*) methods have been used to analyze crimes in many studies. They have contributed to crime and crime fighting institutions with artificial intelligence models. Among these are determining the areas where the crime can be committed and its story (*Hassan & Rahman, 2017*), predicting the crime using spatial, temporal and demographic data (*Zhao & Tang, 2017*), and analyzing crime with literacy, unemployment and development index data (*Vineeth, Pandey & Pradhan, 2016*).

Time series of crime data in San Francisco, Chicago and Philadelphia were used for predicting crimes in the following years. Decision Tree (DT) classification model performed better than K-nearest neighbors (KNN) and Naive Bayes (NB) (*Feng et al., 2018*). Crime data were passed through two different data processing procedures in

Canada. A crime prediction was made with an accuracy between 39% and 44% using the KNN and DTs (*Kim et al., 2018*). Data such as location, type, date, time, latitude and longitude of the crimes taking place in the USA were used as input. The result of the crime predictions made with KNN Classification, Logistic Regression (LR), DTs, Random Forest (RF), Support Vector Machine (SVM) and Bayesian methods was that the KNN classification was the most successful with an accuracy of 78.9% (*Bharati & Sarvanaguru, 2018*).

Thirty-nine different categories of crime data happened in the city of San Francisco were used. By using Gradient Boosted Trees and SVMs, a model dividing crimes into two classes, blue/white collar crime and violent/non-violent crime, was created. High accuracy was achieved in the classification of blue- and white-collar crimes. However, the study did not yield significant results in the classification of violent and non-violent crimes (*Chandrasekar, Raj & Kumar, 2015*). Data from a 10-year murder occurred in Brazil were used. Predictions with 97% accuracy were made using the RF method in order to measure the effect of non-Gaussian residuals and urban metrics on murders. The results of this study revealed that unemployment and ignorance were important variables in murder. Also, the order of importance in predicting the crime was determined (*Alves, Ribeiro & Rodrigues, 2018*). In another study, the type, time and location of the crime data were used to predict crime in certain regions in India. The KNN prediction method was used. The method predicted when and where robbery, gambling, accident, violence, murder and kidnapping crimes would occur. It was found to be more successful than a similar study conducted previously (*Kumar et al., 2020*).

Big data and machine-learning framework were implemented by using crime data collected from social media platforms. The data were gathered through Volunteered Geographic Information, web and mobile crime reporting applications. Crime predictions were produced from the collected data using the NB algorithm. The purpose of these predictions is to determine the location of possible crimes and prevent them (*Jha, Jha & Sharma, 2019*). The demographic and geographical data of the events that took place in the previous years were used to predict the terrorist incidents in India. This model predicted terrorist incidents using artificial intelligence algorithms and achieved results with relatively high accuracy (*Verma et al., 2019*).

The cyber-crime data analyzed were publicly available data from social media platforms, including Facebook and Twitter. Researcher compared the algorithms according to the F-measure value, which is the degree of accuracy and precision. With an accuracy of 80%, RF algorithm was found the best fit in the scenario. Threats were automatically determined through the model analyzing cyber-crimes (*Arora, Sharma & Khatri, 2019*). Real-time crime data published in the internet news were used through the screening program. SVM, Multinomial NB and RF classification methods were used. The data were separated into two; crime and non-crime. The most important part is that it currently provides analysis to the news (*Ghankutkar et al., 2019*). Cyber-crime incident data occurring in India were classified using machine-learning techniques. The model, which predicted crimes with 99% of accuracy, reduced amount of time spent in analysis and manual reporting (*Ch et al., 2020*). A universally compared intrusion detection dataset obtained from Kaggle was used. DNN was used to develop an effective and efficient intrusion detection system in the

IoMT environment to classify and predict unforeseen cyber-attacks. DNN outperformed machine-learning algorithms (*Swarna Priya et al., 2020*). When the related works are examined, a summary is presented in Table 1.

Based on the review of the literature, we can suggest that cyber-attacks and crimes are important to study since they cause substantial damages to individuals and states. As we could observe the fields where machine-learning methods were used and these methods were successful in predictions, they were used in our study. Studies made significant contributions to the literature and particularly to the criminal units conducting investigations. In these studies, general crimes, cyber-crimes and attacks are generally used as a dataset. The actual dataset based on personal attributes is studied to a lower extent and therefore a machine learning-based model using the dataset of our study is proposed. Cyber-attack and perpetrator estimation method is tackled due to the importance of fields that are studied.

## MATERIALS AND METHODS

When people become a victim of a crime, they resort to cops dealing with that particular type of crime. These data are recorded in detail in the database of this unit. Police units report these crimes by their type, method, year, etc. They prepare statistics according to their characteristics, analyze and visualize them. When multiple cyber-attacks are made simultaneously against a victim, this is recorded in police records as a single attack. Therefore, it is necessary to look at the details of the event, rather than the statistical data, in order to understand whether multiple methods are used. Although a large number of crimes exist in the database, the focus is cyber-crimes as in recent years. Cyber-crimes have caused considerable material and moral damages and they have not been prevented yet. Cyber-crime is chosen as the subject matter since most studies on this field do not employ actual data. In the proposed model, the objective is to take preventive measures against future crimes based on the characteristics of the victim. In addition, it will provide advantages to the police department in predicting cyber-crimes, profiling these crimes, perpetrators and victims. In addition, thanks to the model, consequential suffering will be prevented. The outcome of the study will enable to take tailored measures and facilitate informing people of the crimes they may be exposed to. Our dataset was real cyber-crime data that occurred in the province of Elazığ between 2015 and 2019. Accessing to actual data and preparing these data for processing with machine-learning methods was a challenging process. When the dataset was obtained, all cyber-crime details were examined. The redundant areas were removed using various data science methods. The number of crimes, damages, attacks and methods of attack in the dataset are shown in Fig. 1. In addition, the details of these four features are divided into color groups. With these data, predictions were made using various libraries in the Python 3.7 program. Main libraries of this program such as Numpy, Pandas, Matplotlib were used and the data were visualized through this program. The key advantages of using machine-learning methods in the paper are; the possibility to recognize multiple structured and unstructured data patterns, high-level success in detecting crime tactics as they change, extraction

**Table 1 Related work summary comparison.**

| Author | Year | Country | Dataset | Features | Method | Accuracy | Advantage |
|---|---|---|---|---|---|---|---|
| Chandrasekar et al. | 2015 | USA | General Crime in San Francisco | Date, Day of the week, Name of the Police Department District, Address, Latitude, Longitude, Category, Description, Resolution | RF, SVM, Gboost | Md1 SVM 96% Md1 Gboosted 75.02% | Including time series modeling to understand temporal correlations of blue/white collar and Violent/Nonviolent crime classification data and predicting fluctuations in different crime categories |
| Feng et al. | 2018 | USA | General Crime in San Francisco, Chicago, Philadelphia | Date, Category, Descript, Dayof Week, P. D. District, Resolution, Address, x, y, Dome, Arrest | NB, KNN, RF, XGBoost | XGboost 70% | Predicting overall crime in the coming years using time series |
| Kim et al. | 2018 | Canada | General Crime in Vancouver | Crime Type, Month, Day, Hour, Weekday, Address, Neighborhood, x, y | KNN, XGBoost | Md1 Gboost 41.9% Md2 Gboost 43.2% | Predicting crime location |
| Bharati et al. | 2018 | USA | General Crime in Chicago | Block, Location, District, Community Area, Latitude, X, Y, Longitude, Hour, Month, | KNN, LR, DT, RF, SVM, NB | KNN 78.9% | Detecting, predicting, and solving crimes much faster, thereby lowering the crime rate |
| Alves et al. | 2018 | Brazil | Murder Crimes | Child labor, Elderly -Female- Male Population, GDP, Homicides, Income, Literacy, Sanitation, Suicides, Traffic Accidents, Unemployment | RF | RF 97% | The prediction revealed that unemployment and ignorance were the most important variables in defining murders in Brazilian cities, and the order of importance of urban indicators in predicting crimes was determined. |
| Verma et al. | 2019 | India | Terrorist Incidents | Incident Location, Time, Type, Weapon, Intensity of Attack, Capital, Perpetrator | DT, RF, IG | RF 84% | Predicting the occurrence of terrorist incidents on demographic and geographic data |
| Jha et al. | 2019 | India | Murder Crimes on the Internet | Crime Type, Year, Address, Place, Territory | NB | NB 51.2% | Analyzing crimes in minimum time and predicting the location and type of future crimes |
| Arora et al. | 2019 | Public | Cybercrime Data Publicly Available on Social Media | Synonyms, Age, Location, Gender, Hashtags, and Sarcasm | RF | RF 80% | Detecting cyber threats automatically |

| Author | Year | Country | Dataset | Features | Method | Accuracy | Advantage |
|---|---|---|---|---|---|---|---|
| Ghankutkar et al. | 2019 | Public | Real-time Crime Data from HuffPost News Site | Category, Headline, Authors, Link, Short Description, Date | SVM, MNB, RF | RF 85.83% | Providing analysis to current news after being classified as crime and non-crime data |
| Kumar et al. | 2020 | India | Murder, Kidnapping, Violence, Robbery, Gambling, Accident, Indore, Crimes | Hour, Longitude, Latitude, Day of the Week, Week, Month | KNN | Prev 93.23% Proposed 99.51% | Predicting the type of crime, and where and when it may occur |
| Ch et al. | 2020 | India | Cybercrime | Incident, Harm, Access, Year, Violation, Victim, Offender | SVML, LR MNB, RF | LR 99.3% | Finding and analyzing cyber-attacks that exploit vulnerabilities. |

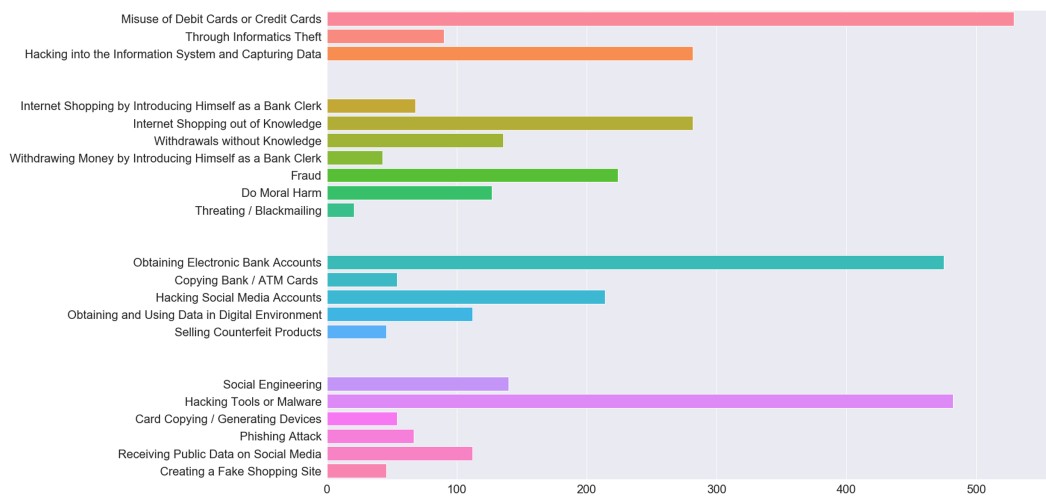

**Figure 1 The number of crimes, damages, attacks and methods of attack in the dataset.**

of relationships between complex data and the ability to produce results unpredictable by humans.

## Features

Feature selection is the process of selecting interrelated and useful features in the dataset. It saves time and memory during data training in machine learning. If the features are improperly selected, time required for training may increase. This makes interpretation of the model difficult and increases the error rate in the model. The attributes of the features in our dataset were determined. Each entry was related to a real crime shown in Table 2. These features were used as inputs in training data. In Fig. 2, the criteria of the features in our dataset are determined as median, maximum and minimum.

**Table 2 The attributes used for feature selection.**

| | |
|---|---|
| Crime | • Misuse of Debit Cards or Credit Cards<br>• Through Informatics Theft<br>• Hacking into the Information System and Capturing Data |
| Gender | M – F |
| Age | 27 and under/Between 28 and 37/Between 38 and 50/51 and above |
| Income | Low – Middle – High |
| Job | Other – Student – Retired – Justice and Security – Health Sector Manager – Housewife – Education – Technical – Finance Sector |
| Marital status | Single – Married |
| Education | Primary Education – High School – Undergraduate – Graduate |
| Harm | • Internet Shopping by Introducing Himself as a Bank Clerk<br>• Internet Shopping out of Knowledge<br>• Withdrawals without Knowledge<br>• Withdrawing Money by Introducing Himself as a Bank Clerk<br>• Fraud<br>• Do Moral Harm<br>• Threating/Blackmailing |
| Attack | • Obtaining Electronic Bank Accounts<br>• Copying Bank/ATM Cards<br>• Hacking Social Media Accounts<br>• Obtaining and Using Data in Digital Environment<br>• Selling Counterfeit Products |
| Attack method | • Social Engineering<br>• Hacking Tools or Malware<br>• Card Copying/Generating Devices<br>• Phishing Attack<br>• Receiving Public Data on Social Media<br>• Creating a Fake Shopping Site |
| Perpetrator | Known – Unknown |

## Preprocessing

Standardization is the rescaling of features in a normal distribution. This needs to be completed before using machine-learning methods. The data were made suitable, numbers from 1 to 10 were given according to the variety of data in the columns. The StandardScaler () in the Python library was used to optimize the data to be used in algorithms. The bidirectional relationship between type of crime and damages, attack and method of attack is shown in Fig. 3. The data were divided into 80% training data and 20% test data.

In the first model, the method of attack was tried to be predicted by giving the features of crime, gender, age, job, income, marital status, education, attack, harm and perpetrator.

In the second model, the perpetrator of crime was tried to be predicted by giving the features of crime, gender, age, income, job, marital status, education, attack, harm and attack method.

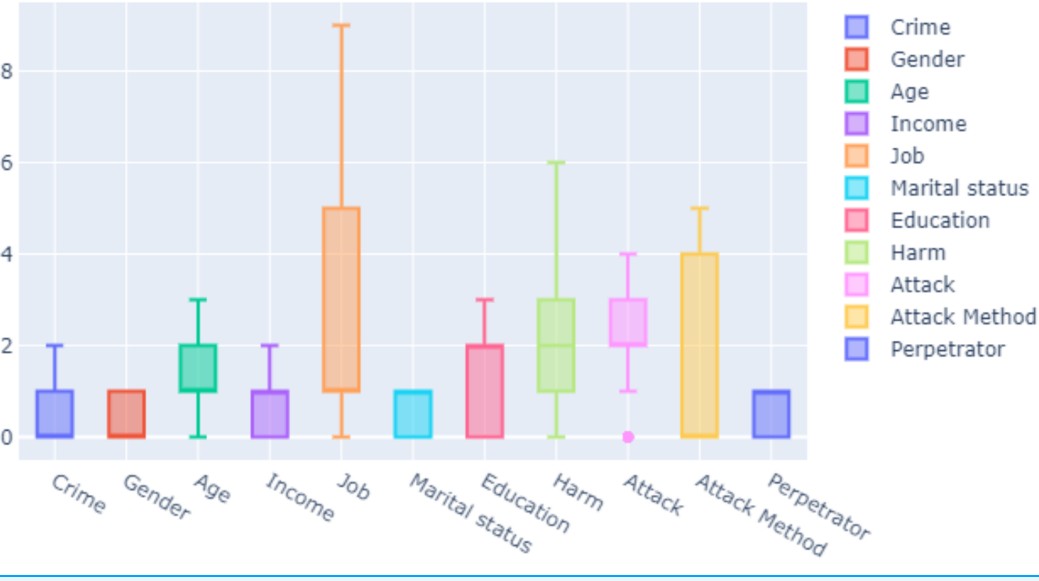

**Figure 2** Measure of features.

## Logistic regression

It is an equation allowing one to predict the value of one from the other, based on the relationship between two or more variables. Assuming that the variance of $y$ is constant, the coefficients can be solved using the least squares method. This minimizes the error between the actual data point and the regression line. The equation for the LR line is given as:

$$y' = b + w_1 x_1 + w_1 x_1 + \ldots + w_n x_n \tag{1}$$

Where,
$y'$ is the desired output,
$b$ is bias value,
$x$ is the property of the input,
$w$ is the weight of the features.

## K-nearest neighbors

The KNN classifier is based on the distance function that measures the difference and similarity between two or more samples. The Euclidean distance $d(x, y)$ between two samples is defined as:

$$d(x, y) = \sum_{k=1}^{n} (x_k + y_k)^2 \tag{2}$$

Where,
$x_k$ and $y_k$ are the $n^{th}$ element.
$n$ is the $n^{th}$ property of the dataset.
First, the $k$ parameter is determined and the distance of the new data to be included in the dataset is calculated one by one according to the existing data. The closest neighbour is found and assigned to the neighbour class $k$.

# PeerJ Computer Science

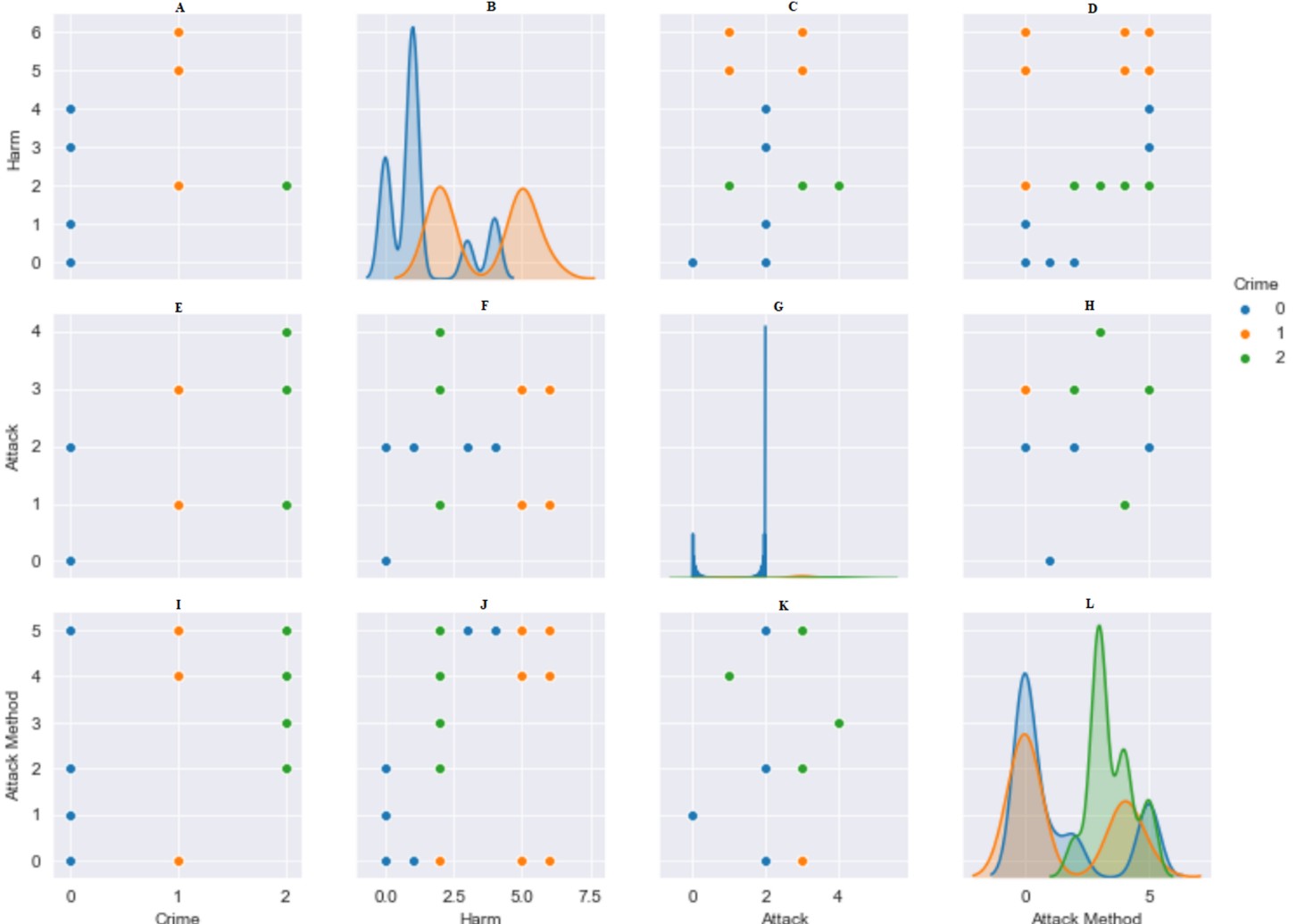

**Figure 3 Pairwise relationship between crime, harm, attack and attack method.** (A) Harm: Crime. (B) Harm: Harm. (C) Harm: Attack. (D) Harm: Attack Method. (E) Attack: Crime. (F) Attack: Harm. (G) Attack: Attack. (H) Attack: Attack Method. (I) Attack Method: Crime. (J) Attack Method: Harm. (K) Attack Method: Attack. (L) Attack Method: Attack Method.

## Support vector machine

This method includes support vector classification and support vector regression. SVM is based on the concept of decision limits, supporting both binary and more classifications. Considering the training data $D$:

$$D = \{(x_i, y_i) \mid x_i \in R\,p, y_i \in \{-1, 1\}\}_{i=1}^{n} \qquad (3)$$

Here $y_i$ is 1 or $-1$ and determines the class to which the $x_i$ point belongs. Every $x_i$ is a $p$-dimensional real vector. The support vector is at the closest point to the hyperplane of optimum separation. In the classification process, mapping input vectors on the separation hyperplane side of the feature space fall into one class, and locations fall into the class on the other side of the plane.

## Naive Bayes

The NB classifier is a simple probability classifier based on applying Bayes' theorem with strong independence assumptions between properties. A NB based on the multi-label classification model can be applied. Given a dataset $\{a_1, a_2,\ldots, a_j\}$ labelled $\{v_1, v_2,\ldots, v_j\}$, the results are predicted using the following equation:

$$v_{NB} = \frac{\arg\max}{v_j \in V} P(v_j) \prod_i Pa_i|v_j \tag{4}$$

## Decision tree

Decision tree is a classification method that creates a tree-like model consisting of decision nodes and leaf nodes by classification, feature, and target. A DT or classification tree is a tree where each internal node is labelled with an input property. Arcs from this tagged node are tagged with each of the possible values of the target attribute or lead to a sub-decision node in a different input attribute. A tree can learn by dividing the set of resources into subsets according to an attribute value test. This process is repeated in a recursive manner, called recursive partitioning, on each derived subset. The iteration is complete when the subset in a node has the full value of the target variable or the division no longer adds value to the predictions.

$$(x, \ Y) = (x_1, \ x_2, \ x_3, \ \ldots, \ x_k, \ Y) \tag{5}$$

The dependent variable $Y$ is the target variable that we are trying to understand, classify or generalize. The vector $x$ consists of input variables such as $x_1$, $x_2$, $x_3$ used for this task.

## Random forest

Random forest is an algorithm that creates classifier on training data and combines outputs to make the best predictions on test data. Randomness minimizes variance to avoid over-learning according to training data.

$$y = \arg\max_{p \,\in\{h(x_1)..h(x_k)\}} \left\{ \sum_{j=1}^{k} \left( I\left( h\left( x|\theta_j \right) = p \right) \right) \right\} \tag{6}$$

Where,
$h(x|\theta)$ is a classification tree,
$k$ is the number of the tree selected from a pattern random vector.
If $D(x, y)$ represents the training data, each classification tree in the ensemble is constructed using a different subset $D_{\theta_k}(x, \ y) \subset D(x,y)$ of the training data. Each tree then works like normal DTs. Data is segmented by randomly selected value until it is fully partitioned, or the maximum allowed depth is reached.

## eXtreme gradient boosting

The first step in eXtreme Gradient Boosting (XGBoost) is to make the first prediction (base score). This prediction can be any number, as the correct result will be reached by converging with the operations to be done in the next steps. This number is 0.5 by default.

First, the loss function $L(y_i, F(x))$ is created. $y_i$ is the observed value, $F(x)$ is the predicted value:

$$F_0(x) = \arg \min_{\gamma} \sum_{i=1}^{n} (L(y_i, \gamma)) \tag{7}$$

Here, the constant variable is determined. It is the value loss function in sigma in the formula. $\gamma$ (gamma) is the predicted value.

$$r_{im} = -\left[ \frac{\partial L(y_i, F(x_i))}{\partial F(x_i)} \right]_{F(x)=F_{m-1}(x)} \quad i = 1, \ldots, n \tag{8}$$

Where,

$r$ means residual,

$i$ is observation number,

$m$ denotes the number of the established tree.

The basic learning terminal node in tree growth is the regression tree. It is formulated below:

$$\gamma_{im} = \arg \min_{\gamma} \sum_{x_i \ R_{ij}} L(y_i, F_{m-1}(x_i) + \gamma) \qquad j = 1 \ldots J_m \tag{9}$$

$$F_m(x) = F_{m-1}(x) + \sum_{j=1}^{j_m} \gamma_{jm} I(x \in R_{jm}) \tag{10}$$

## Accuracy, precision, recall, *F*1-score

Accuracy (Acc) score is a method used to evaluate the performance of the model made by comparing the predictions made after running the algorithm with the test data. A value between 0 and 1 is produced according to the ratio of the entire predicted value for a prediction to match with the real values. To determine the accuracy of the forecast:

- TP = Prediction is positive(normal) and actual is positive(normal).
- FP = Prediction is positive(normal) and actual is negative(abnormal).
- FN = Prediction is negative(abnormal) and actual is positive(normal).
- TN = Prediction is negative(abnormal) and actual is negative(abnormal)

The other evaluation metrics for the proposed model are precision, recall and *F*1-score. Precision (P) is the rate of correctly classified positive instances to the total number of positive instances. Recall (R) shows how successfully positive instances are predicted. *F*1-Score (*F*1) is the weighted average of the Precision and Recall values.

$$Acc = \frac{TP + TN}{TP + TN + FP + FN} \tag{11}$$

$$P = \frac{TP}{TP + FP} \tag{12}$$

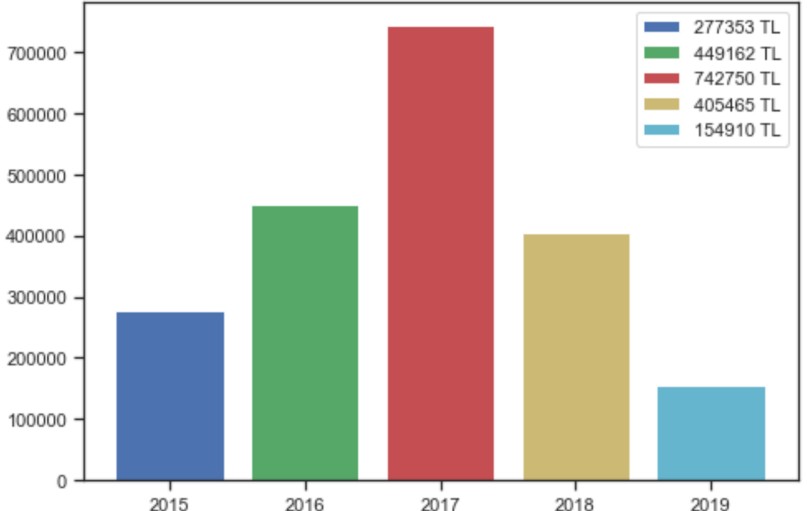

**Figure 4 Damage caused by cyber-attacks in Elazığ province.**

$$R = \frac{TP}{TP + FN} \tag{13}$$

$$F1 = \frac{2TP}{2TP + FN + FP} \tag{14}$$

## RESULTS AND DISCUSSION

The study aims to analyze the data collected about incidents correctly, to avoid crimes and to catch the perpetrators. The main subject of this paper is to draw conclusions from the analyzed data and combat crimes based on the outcome. These results will reveal and shed light on the investigations carried out by law enforcement officers and any concealed facts. Based on the information on the victim and the method of the cyber-crime, and whether the perpetrator is identified or not, machine-learning methods may be used to determine if the same perpetrator carried out the cyber-attack. The damages suffered by the victims in cyber incidents in Elazığ province have been discovered over the years through various methods. The sum of monetary damages suffered by each victim in the dataset was obtained by summing over the years. It is thought that the decrease in such incidents, observed especially after 2017, results from deterrence secured by the laws and awareness activities. The amount of economic losses due to cyber-attacks is profoundly serious in Elazığ, as shown in Fig. 4. The damage mentioned above is enough to show the importance of dealing with cyber security and attack methods.

In this section, results obtained by use of SVM (Linear), RF, Logistic Regression, XGBoost, SVM (Kernel), DT, KNN, NB algorithms are presented. We can evaluate the Pearson correlation coefficient between these data as shown in Fig. 5. This correlation matrix shows that there are substantial correlations between practically all pairs of variables.

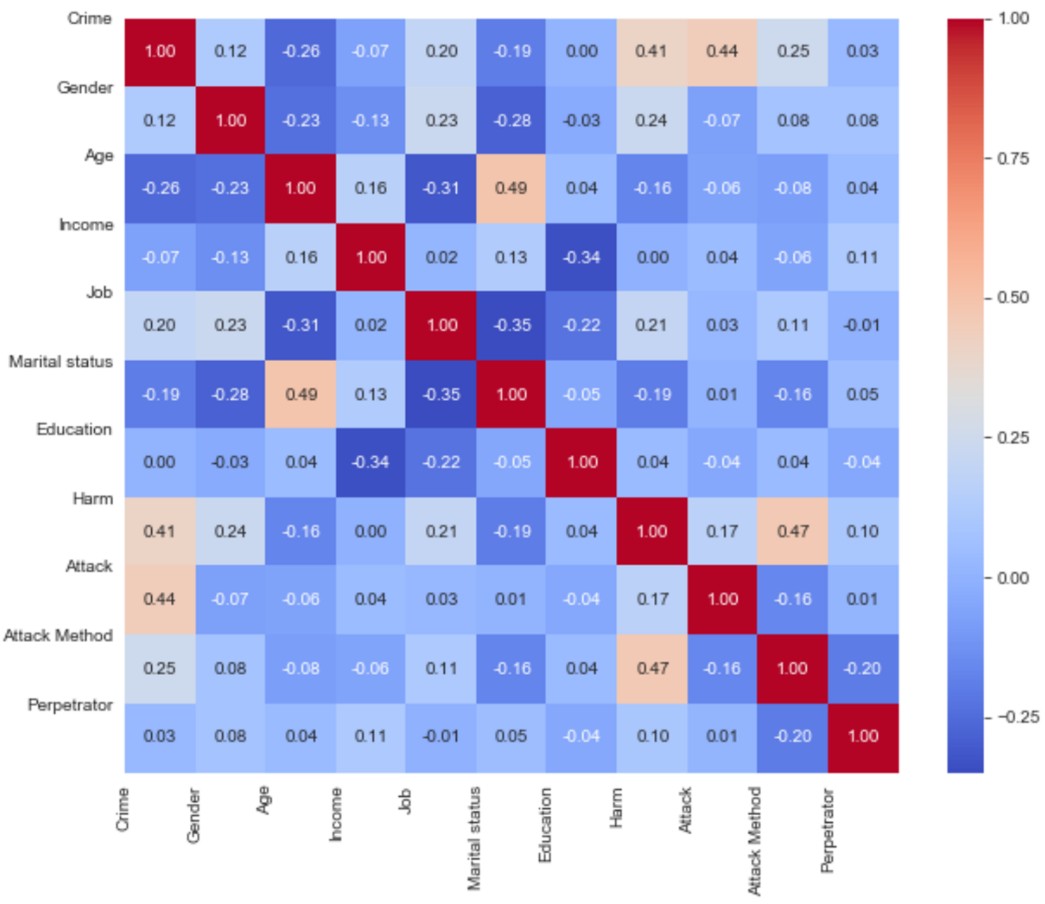

**Figure 5 Pearson's linear correlations matrix.**

**Table 3 Performance of machine learning algorithms for model 1.**

|  | Accuracy % | Precision % | Recall % | F1-score % |
|---|---|---|---|---|
| LR | 93.92 | 94.41 | 93.92 | 94.1 |
| KNN | 89.5 | 71.73 | 76.24 | 72.69 |
| SVML | 95.02 | 95.43 | 95.03 | 95.16 |
| SVMK | 92.82 | 92.99 | 92.82 | 92.88 |
| NB | 81.77 | 81.79 | 81.77 | 81.23 |
| DT | 92.26 | 92.3 | 92.27 | 92.28 |
| RF | 94.48 | 94.48 | 94.48 | 94.48 |
| XGBOOST | 93.36 | 92.75 | 92.82 | 92.76 |

During the experiment, the dataset was first trained and tested in all algorithms. Accuracy and evaluation criteria were also adopted. Accuracy, precision, recall and $F1$ score values were obtained by comparing the predicted values with the test data.

The prediction accuracy, precision, recall and $F1$-score data among the algorithms of the first model predicting the method of attack are shown in Table 3. The comparison of accuracy is shown in Fig. 6. When the results were compared, SVML showed the best

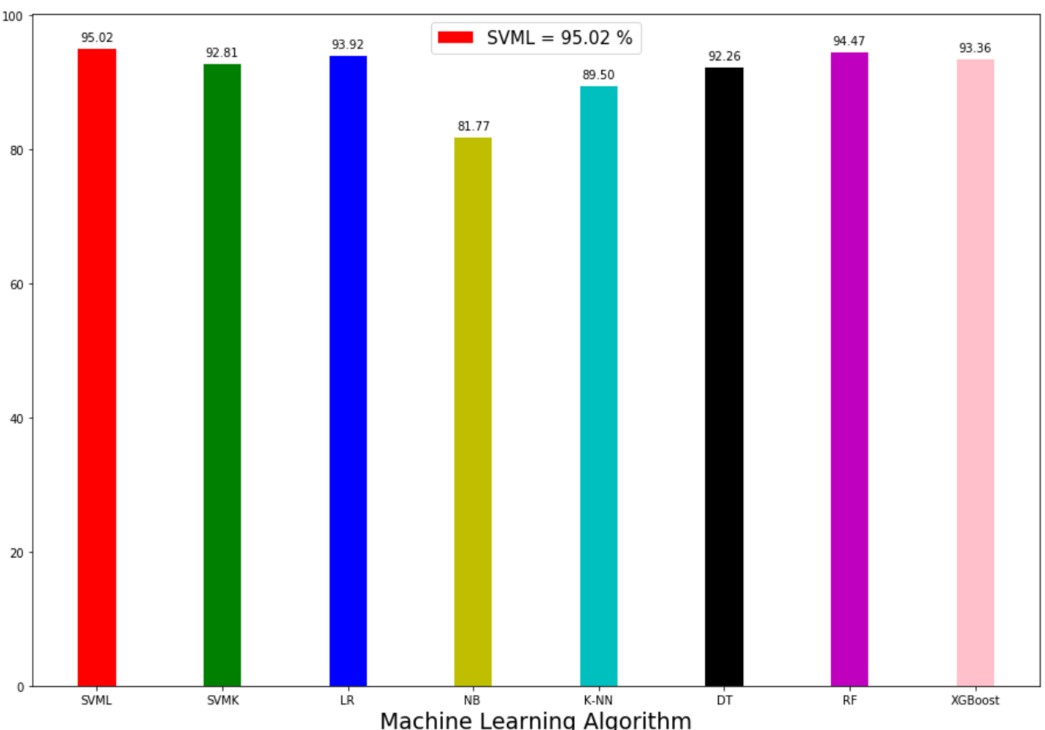

**Figure 6 Accuracy rate of algorithms applied in attack method prediction.**

performance with an accuracy rate of 95.02% in terms of prediction accuracy. SVML algorithm outperformed RF, LR, XGBoost, SVMK, DT, KNN and NB algorithms by 0.54%, 1.1%, 1.66%, 2.2%, 2.76%, 5.52% and 13.25% margin, respectively. NB showed the poorest performance with a rate of 81.7%. The performances of algorithms other than NB were close to each other. The distribution graph of the actual values and the predicted values obtained by the SVML algorithm are shown in Fig. 7A and the error matrix is shown in Fig. 7B.

Comparing the model in terms of precision, recall and $F$1-scores, the best result was also obtained with SVML algorithm, albeit a small margin. While LR, SVMK, DT, RF and XGBoost gave results above 92%, their performances were close to each other. Out of these three scores, a difference of approximately 10% was found between the underperforming KNN and NB. In general, all algorithms produced sufficient results. These results showed that the machine-learning approach could be successfully applied to predict the cyber-attack method. In the model to be created, when the features of a person (Table 2) are entered, it can be predicted which crime such person will be exposed to. Also, warning systems can be created for groups of persons.

The prediction accuracy, precision, recall and $F$1-scores data among the algorithms of the second model that predicts the attacker are given in Table 4. The comparison of accuracy is shown in Fig. 8. LR showed the best performance in this model with 65.42% and SVML, KNN, SVMK, XGBoost, RF, DT and NB algorithms achieved accuracy with a margin of 0.78%, 0.85%, 1.33%, 1.39%, 1.88%, 2.44% and 3.34%, respectively. Even

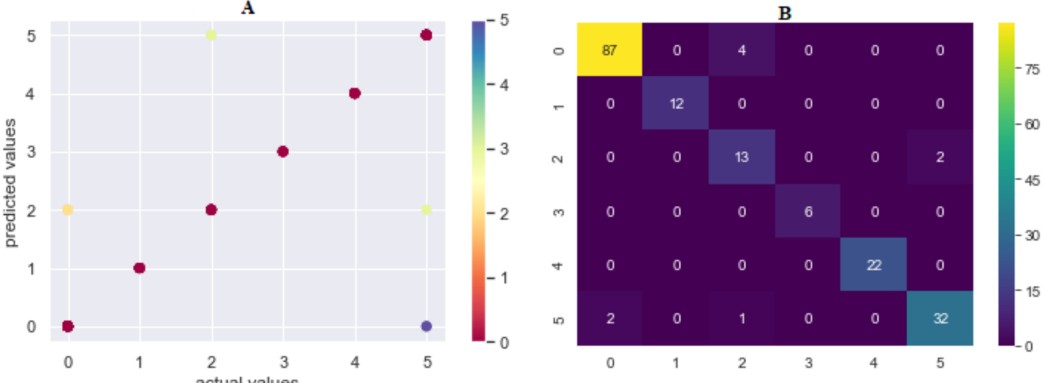

**Figure 7** (A) Model 1 comparison values (results are become more red as they approach the actual values and more purple as they move away); (B) confusion matrix of predicted values. Where 0 is "Hacking Tools or Malware", 1 is "Card Copying/Generating Devices", 2 is "Phishing Attack", 3 is "Creating a Fake Shopping Site", 4 is "Receiving Public Data on Social Media", 5 is "Social Engineering".

**Table 4** Performance of machine learning algorithms for model 2.

|  | Accuracy % | Precision % | Recall % | F1-score % |
|---|---|---|---|---|
| LR | 65.42 | 60.67 | 60.22 | 59.14 |
| KNN | 64.57 | 56.85 | 56.91 | 56.85 |
| SVML | 64.64 | 65.75 | 64.64 | 63.54 |
| SVMK | 64.09 | 65.24 | 64.09 | 62.90 |
| NB | 62.08 | 57.93 | 57.46 | 55.66 |
| DT | 62.98 | 63.50 | 62.98 | 62.18 |
| RF | 63.54 | 63.91 | 63.54 | 62.92 |
| XGBOOST | 64.03 | 65.74 | 65.19 | 64.55 |

though NB showed the poorest performance, the performances of algorithms were very close to each other. The distribution graph of the actual values and the predicted values obtained by the SVML algorithm are shown in Fig. 9A and the error matrix is shown in Fig. 9B.

In terms of precision, recall and F1-scores, the results of the algorithms varied between 55% and 65%. The results were not satisfactory. Based on the known/unknown feature of the perpetrator, we aimed to find out whether the same perpetrator committed the crime after comparing the features of the attacker who carried out the incident. However, the results of the model indicated that a new model should be created by adding new attributes.

When the papers shown in the table in "Related Works" are compared with our model, eight of the datasets of these works include general street-crimes. One of them works with terrorist crimes, and two of them work with cyber-crimes. Cyber-crime data is one of the less studied crime types in the literature so that we use these data for our proposed study.
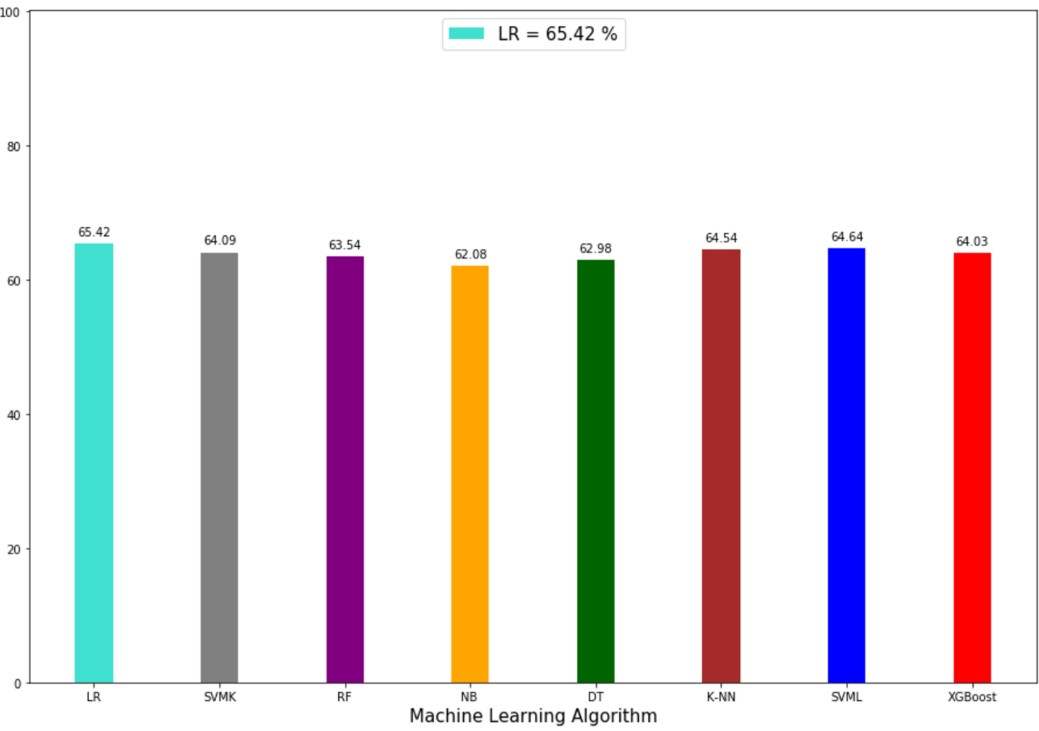

**Figure 8 Accuracy rate of algorithms applied in perpetrator prediction method.**

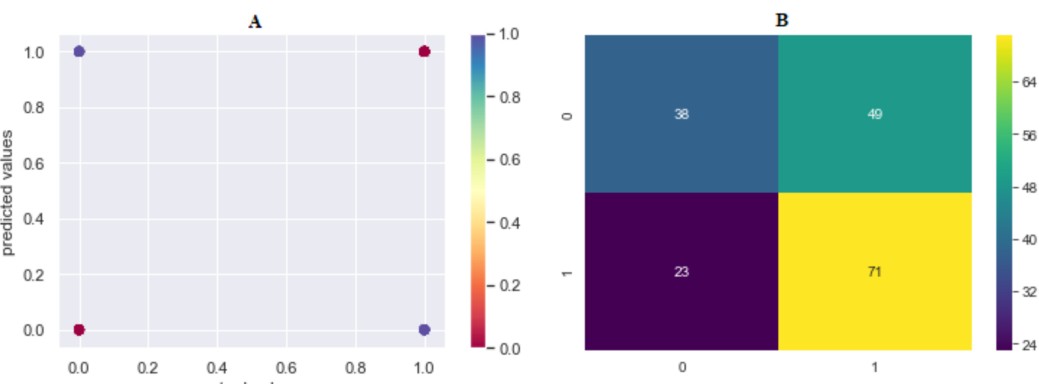

**Figure 9 (A) Model 2 comparison values (results are become more red as they approach the actual values and more purple as they move away); (B) confusion matrix of predicted values.** Where 0 is "Perpetrator Known", 1 is "Perpetrator Unknown".

In the other predictions, it is seen that the data such as the type, time, place, region, attacker, demographic and regional factors of the crime are mostly used as features. In our study, predictions are made according to the features of age, gender, income, occupation, harm, and attack methods because there are very few studies using these features.

When the studies conducted by *Arora, Sharma & Khatri (2019)* and *Ch et al. (2020)* were compared with our study, it was found out that some features were similar. All studies focused on cyber-crime, the datasets and parameters were different from each other,

though. The accuracy rate was found to be 80% in the study conducted by *Arora, Sharma & Khatri (2019)* and 99.3% in the study carried out by *Ch et al. (2020)*.

The limitation of our study is the quantity of data set since our data set consists of actual data. The temporal data enables estimation of time series; however, these data are not available in our dataset. Likewise, if the technical details of the attacks were available in the police records, detailed estimates could make it easier to catch the perpetrator.

Many studies have predicted where and when crimes will occur in the future. However, many of them have not touched upon the method by which crimes occur, how they can be prevented, and what the features of the perpetrator are. One of the key advantages of this study is using actual data and it is a preliminary step towards profiling for people having similar features with the attacked victims. Another advantage of the proposed study is predicting what the cyber-attack method will be and whether its perpetrator can be detected. Our results show that any exposure to cyber-crimes reduces as the level of education and income increases.

## CONCLUSIONS AND FUTURE WORK

This paper suggests a method that predicts and detects cyber-attacks by using both machine-learning algorithms and the data from previous cyber-crime cases. In the model, the characteristics of the people who may be attacked and which methods of attack they may be exposed to are predicted. It has been observed that machine-learning methods are successful enough. The SVMs linear method is the most successful of these methods. The success rate of predicting the attacker who will make a cyber-attack in the model is around 60%. Other artificial intelligence methods may be able to try to increase this ratio. In our approach, it is concluded that it is necessary to draw attention to especially malware and social engineering attacks. It was found that the higher the levels of the victim's education and income are, the less the probability of cyber-attack is. The primary focus of this study is to lead law enforcement agencies in the fight against cyber-crime and provide faster and more effective solutions in detecting crime and criminals. New training and warning systems can be created for people with similar characteristics by the evaluation of the characteristics of the attack victims emerged in our analysis study.

For future works; crime, criminal, victim profiling and cyber-attacks can be predicted using deep learning algorithms and the results can be compared. Based on the talks with other authorized units having crime databases, cyber-crime data of other provinces may also be obtained to use for comparison with this study. The data of other provinces can be compared to similar studies. Intelligent criminal-victim detection systems that can be useful to law enforcement agencies in the fight against crime and criminals can be created to reduce crime rates.

### Funding
The authors received no funding for this work.

## Competing Interests

The authors declare that they have no competing interests.

## Author Contributions

- Abdulkadir Bilen conceived and designed the experiments, performed the experiments, analyzed the data, performed the computation work, prepared figures and/or tables, authored or reviewed drafts of the paper, and approved the final draft.
- Ahmet Bedri Özer conceived and designed the experiments, performed the experiments, analyzed the data, performed the computation work, prepared figures and/or tables, authored or reviewed drafts of the paper, and approved the final draft.

## Data Availability

   The raw data and the code are available in the Supplemental Files.

## Supplemental Information

Supplemental information for this article can be found online at http://dx.doi.org/10.7717/peerj-cs.475#supplemental-information.

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
