# Peer review of "Cyber-attack method and perpetrator prediction using machine learning algorithms"

_PeerJ Computer Science, doi:10.7717/peerj-cs.475_

## Round 0.1 · original submission · Major Revisions

· Academic Editor

Major Revisions

The reviewers' concerns are highly relevant and should be carefully addressed.

Reviewer 1 ·

Basic reporting

The paper utilises different Machine Learning techniques to predict the attack method and attacker. A dataset from of attacks (2015 - 2019) is used.
The paper in its current form suffers from these shortcomings that the authors should consider.

The paper is well written and easy to follow.
The English language needs some minor revisions and edits.
The authors provide their code and data.

1- The related work is missing context and insights. This section is a group of relevant papers, however, there is no link or discussion between them.
The authors report multiple studies that do similar work, however, no insights are reported. The authors are advised to give the reader some context to understand their approach afterwards.

2- In the abstract, the authors should stress on the problem and what they are trying to solve, how they solve it, etc. The dataset should also be mentioned. In line 25, SVM with a linear kernel is said to be the best, however, the authors do not provide any discussion about this. What exactly is being predicted.


3- Table 2 should be moved to the related work section.

4- It is advised that the authors be more precise when reporting other people research. What dataset do they use? What are they predicting/detecting? etc.

5- Some references are old. For example in line 36, a recent supporting reference about UK and US figures is needed.
Lines 70-73 require a supporting reference. The paper claims that social engineering attacks are the most difficult. Is this based on their findings? Is it in one place or is this a global finding?

6- In Figure 1, are these insights based on the dataset? If so, the authors are advised to make this clear. Also, why is there a peak in 2017? Were measures put in place to reduce the damage in 2018 and 2019?

Experimental design

1- The aim and objective of this paper are vague. I recommend that the authors clarify the problem they are tackling. Also, who will this model benefit and how? Placing this paper in the wider research context is required.

2- The dataset is comparably small (~900 records). Is there any way the validate the model on a larger dataset? Also, I was expecting validating the model on a benchmark dataset.

3- The authors are advised to add a description of how the dataset is generated.

4- In the dataset, did the authors consider overlapping attacks? what about cases where multiple methods are used? How is this handled?

5- The authors provide the background of each of the algorithms they use, however, no discussion/investigation of the results is provided. Not enough details of the experiments are mentioned in the paper.

Validity of the findings

1- The paper barely discusses the results. The authors are advised to provide more results and comment on them. Also, the limitations of the work are not mentioned in the paper. I recommend adding a section for limitations and discussion.

2- The authors claim that they predict the perpetrator, however, the dataset comprises 'Known/Unknown' categories. how is this prediction evaluated?

3- Label encoder is good when encoding ordinal data, however, since some of the columns comprise nominal data, One-Hot encoding is better to use in this case.
The year should be dropped from Table 1, the authors do not use it.

4- In lines 269-282, how are the results comparable? The paper says that its performance is better than seven studies. However, the datasets are not the same and the parameters are different, how is the comparison conducted The authors are advised to recheck this and validate their findings.

Additional comments

1- In lines 59 - 62, the authors should refer to the sections and not chapters.

2- The authors do not consider any discusses IoT attacks in section I.

3- In line 92, it should be 78.9\%.

4- In Table 2, it would be beneficial to add the year and the country. Also, adding features relations would add to this table.

Reviewer 2 ·

Basic reporting

In this paper, the analyze cyber-crime in two different models with machine-learning methods and predict what effect the determined features have on the detection of cyber-attack method and the perpetrator. In their approach, it is seen that the Support Vector Machine Linear is the most successful in the cyber-attack method with 95% of accuracy. The Logistic Regression is the leading method in detecting the attacker with a rate of 65.4% of accuracy. The results reveal that the higher the level of victim’s education and income are, the less the probability of cyber-attack is.

Experimental design

The Support Vector Machine Linear is the most successful in the cyber-attack method with 95% of accuracy. The Logistic Regression is the leading method in detecting the attacker with a rate of 65.4% of accuracy. The results reveal that the higher the level of victim’s education and income are, the less the probability of cyber-attack is.

Validity of the findings

Good

Additional comments

1. The main objectives and contributions are to be highlighted in the paper.
2. The related works section should be summarized by the gaps identified in the existing literature and how the authors are going to address them in this paper (atleast one of them).
3. Some of the recent and relevant works such as the following can be discussed in the paper:
a) An effective feature engineering for DNN using hybrid PCA-GWO for intrusion detection in IoMT architecture
b) A Novel PCA-Firefly based XGBoost classification model for Intrusion Detection in Networks using GPU
c) A Review on Cyber Crimes on the Internet of Things
4) The authors should justify why they chose the machine learning algorithms they have used in the paper.

---

## Round 0.2 · Minor Revisions

· Academic Editor

Minor Revisions

.Kindly address the reviewer's remaining concerns.

Reviewer 1 ·

Basic reporting

- I thank the authors for their efforts in modifying the paper and replying to the concerns.
- The authors improved the language of the paper, however, I recommend using shorter sentences and having another round of language revision.

Experimental design

- I thank the authors for reporting other matrices which better demonstrate the performance of their proposed model.
- In section III.B, the authors mention the use of cross-validation with k = 3 and a split of 80% for training and 20% for testing. This is inconsistent, please revise.
- Reporting the training performance doesn't add much to the discussion. The authors should focus on validation and testing performance.

Validity of the findings

- It is unclear which fold is reported in Figure 7 and Figure 9.
- How are the models used simultaneously, the paper doesn't provide a hybrid model or an aggregation method for the eight models. The authors should review this claim.
- The authors should not use label encoder for nominal data

Additional comments

- There is an extra sentence in line 312 "formulas are used."
- I recommend improving the resolution of Figure 1. Also, is there ant relevance to the colours/grouping? This should be added to the text.
- The authors should add the labels (percentages) to Figure 6 and Figure 8 to improve readability.

Reviewer 2 ·

Basic reporting

The authors have addressed all my comments. The paper can be accepted for publication in its current form now.

Experimental design

Satisfactory

Validity of the findings

Satisfactory

Additional comments

The authors have addressed all my comments. The paper can be accepted for publication in its current form now.

---

## Round 0.3 · accepted · Accept

· Academic Editor

Accept

Thank you for addressing reviewers concerns.

Reviewer 1 ·

Basic reporting

The authors replied to my concerns.

Experimental design

The authors replied to my concerns.

Validity of the findings

The authors replied to my concerns.

Additional comments

I thank the authors for their efforts in modifying the paper and replying to the concerns